# VARIATIONAL STRUCTURED ATTENTION NETWORKS FOR DENSE PIXEL-WISE PREDICTION

## ABSTRACT

State-of-the-art performances in dense pixel-wise prediction tasks are obtained with specifically designed convolutional networks. These models often benefit from attention mechanisms that allow better learning of deep representations. Recent works showed the importance of estimating both spatial- and channel-wise attention tensors. In this paper we propose a unified approach to jointly estimate spatial attention maps and channel attention vectors so as to structure the resulting attention tensor. Moreover, we integrate the estimation of the attention within a probabilistic framework, leading to VarIational STructured Attention networks (VISTA-Net). We implement the inference rules within the neural network, thus allowing for joint learning of the probabilistic and the CNN front-end parameters. Importantly, as demonstrated by our extensive empirical evaluation on six large-scale datasets, VISTA-Net outperforms the state-of-the-art in multiple continuous and discrete pixel-level prediction tasks, thus confirming the benefit of structuring the attention tensor and of inferring it within a probabilistic formulation.

## 1 INTRODUCTION

Over the past decade, convolutional neural networks (CNNs) have become the privileged methodology to address computer vision tasks requiring dense pixel-wise prediction, such as semantic segmentation (Chen et al., 2016b; Fu et al., 2019), monocular depth prediction (Liu et al., 2015; Roy & Todorovic, 2016), contour detection (Xu et al., 2017a) and normal surface computation (Eigen et al., 2014). Recent studies provided clear evidence that attention mechanisms (Mnih et al., 2014) within deep networks are undoubtedly a crucial factor in improving the performance (Chen et al., 2016b; Xu et al., 2017a; Fu et al., 2019; Zhan et al., 2018). In particular, previous works demonstrated that deeply learned attentions acting as soft weights to interact with different deep features at each channel (Zhong et al., 2020; Zhang et al., 2018; Song et al., 2020) and at each pixel location (Li et al., 2020a; Johnston & Carneiro, 2020; Tay et al., 2019) permits to improve the pixel-wise prediction accuracy (see Fig.1.a and Fig.1.b). Recently, Fu et al. (2019) proposed the Dual Attention Network (DANet), embedding in a fully convolutional network (FCN) two complementary attention modules, specifically conceived to model separately the semantic dependencies associated to the spatial and to the channel dimensions (Fig.1.c).

Concurrently, other approaches have considered the use of structured attention models integrated within a graph network framework (Zhang et al., 2020; Chen et al., 2019; Xu et al., 2017a), showing the empirical advantage of adopting a graphical model to effectively capture the structured information present in the hidden layers of the neural network and thus enabling the learning of better deep feature representations. Notably, Xu et al. (2017a) first introduced attention-gated conditional random fields (AG-CRFs), a convolutional neural network implementing a probabilistic graphical model that considers attention variables as gates (Minka & Winn, 2009) in order to learn improved deep features and effectively fuse multi-scale information. However, their structured attention model is only learned at the spatial-wise level, while channel-wise dependencies are not considered.

This paper advances the state of the art in dense pixel-wise prediction by proposing a novel approach to learn more effective deep representations by integrating a structured attention model which jointly account for spatial- and channel-level dependencies using an attention tensor (Fig.1.d) within a CRF framework. More precisely, inspired from Xu et al. (2017a) we model the attention as gates. Crucially, we address the question on how to enforce structure within these latent gates, in order to

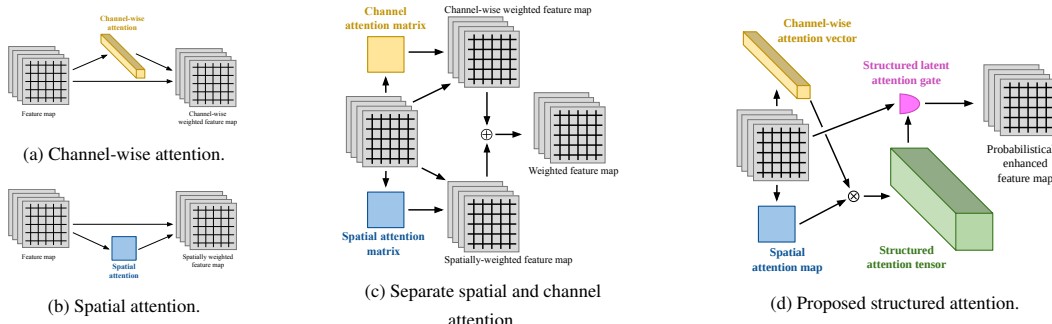

Figure 1: Different attention mechanisms. (a) and (b) correspond to channel-only and spatial-only attention, respectively. (c) corresponds to previous works (Fu et al., 2019) adding ($\oplus$) a channel and a spatial attention tensor. (d) shows the attention mechanism of VISTA-Net: a channel-wise vector and a spatial map are estimated then tensor-multiplied ($\otimes$) yielding a structured attention tensor. The attention tensor acts as a structured latent gate, producing a probabilistically enhanced feature map.

jointly model spatial- and channel-level dependencies while learning deep features. To do so, we hypothesize that the attention tensor is nothing but the sum of $T$ rank-1 tensors, each of them being the tensor product of a spatial attention map and a channel attention vector. This attention tensor is used as a structured latent attention gate, enhancing the feature maps. We cast the inference problem into a maximum-likelihood estimation formulation that is made computationally tractable thanks to a variational approximation. Furthermore, we implement the maximum likelihood update rules within a neural network, so that they can be jointly learned with the preferred CNN front-end. We called our approach based on structured attention and variational inference VarIational STructured Attention Networks or VISTA-Net. We evaluate our method on multiple pixel-wise prediction problems, *i.e.* monocular depth estimation, semantic segmentation and surface normale prediction, considering six publicly available datasets, *i.e.* NYUD-V2 (Silberman et al., 2012), KITTI (Geiger et al., 2013), Pascal-Context (Mottaghi et al., 2014), Pascal VOC2012 (Everingham et al., 2010), Cityscape (Cordts et al., 2016) and ScanNet (Dai et al., 2017). Our results demonstrate that VISTA-Net is able to learn rich deep representations thanks to the proposed structured attention and our probabilistic formulation, outperforming state-of-the-art methods.

**Related Work.** Several works have considered integrating attention models within deep architectures to improve performance in several tasks such as image categorization (Xiao et al., 2015), speech recognition (Chorowski et al., 2015) and machine translation (Vaswani et al., 2017; Kim et al., 2017; Luong et al., 2015). Focusing on pixel-wise prediction, Chen et al. (2016b) first described an attention model to combine multi-scale features learned by a FCN for semantic segmentation. Zhang et al. (2018) designed EncNet, a network equipped with a channel attention mechanism to model global context. Zhao et al. (2018) proposed to account for pixel-wise dependencies introducing relative position information in spatial dimension within the convolutional layers. Huang et al. (2019b) described CCNet, a deep architecture that embeds a criss-cross attention module with the idea of modeling contextual dependencies using sparsely-connected graphs, such as to achieve higher computational efficiency. Fu et al. (2019) proposed to model semantic dependencies associated with spatial and channel dimensions by using two separate attention modules. Zhong et al. (2020) introduced a squeeze-and-attention network (SANet) specialized to pixel-wise prediction that takes into account spatial and channel inter-dependencies in an efficient way.

Attention was first adopted within a CRF framework by Xu et al. (2017a), which introduced gates to control the message passing between latent variables and showed that this strategy is effective for contour detection. Our work significantly departs from these previous approaches, as we introduce a novel structured attention mechanism, jointly handling spatial- and channel-level dependencies within a probabilistic framework. Notably, we also prove that our model can be successfully employed in case of several challenging dense pixel-level prediction tasks. Our work is also closely related to previous studies on dual graph convolutional network (Zhang et al., 2019c) and dynamic graph message passing networks (Zhang et al., 2020), which have been successfully used for pixel-level prediction tasks. However, while they also resort on message passing for learning refined deep feature representations, they lack a probabilistic formulation. Finally, previous studies (Xu et al., 2017c; Arnab et al., 2016; Chen et al., 2019) described CRF-based models for pixel-wise estima-

tion, *e.g.* to learn and optimally fuse deep representations at multiple scales. However, they did not employ structured attention gates.

## 2    VARIATIONAL STRUCTURED ATTENTION NETWORKS: VISTA-NET

As previously discussed, we aim to enhance the learned representation by structuring the attention within a probabilistic formulation. One the one side, inducing structure in the attention mechanisms has been proven to be successful (Fu et al., 2019; Zhong et al., 2020). On the other side, probabilistic formulations combined with deep architectures are interesting for pixel-level prediction tasks (Xu et al., 2017b). Up to our knowledge, we are the first to bring together recent advances in pixel-wise prediction by formulating a novel structured attention mechanism within a probabilistic CRF-like inference framework. Inspired by Fu et al. (2019), where two spatial- and a channel-wise *full-rank* tensors are computed, we opt to infer different spatial and channel attention variables.Very differently from Fu et al. (2019), we propose to structure a generic attention tensor $\mathbf{a}$ of dimension $W \times H \times C$ (widht, height, channels), as the sum of $T$ rank-1 tensors:

$$\mathbf{a} = \sum_{t=1}^{T} \mathbf{m}^t \otimes \mathbf{v}^t, \qquad \mathbf{m}^t \in \mathbb{R}^{1 \times W \times H}, \mathbf{v}^t \in \mathbb{R}^{C \times 1 \times 1}, \tag{1}$$

meaning that $\mathbf{m}^t$ can be understood as an image of $W \times H$ pixels and $\mathbf{v}^t$ as a vector of dimension $C$, and $\otimes$ denotes the tensor product, in the case above leading to a 3-way tensor of dimensions $W \times H \times C$. Each of the tensor products within the sum yields a tensor of rank-1, consequently limiting the rank of $\mathbf{a}$ to be at maximum $T$. Equation (1) is the algebraic expression of the proposed structured attention mechanism, and is the methodological foundation of VISTA-Net.

Moreover, we inspire from the CRF formulation with gating variables proposed in (Xu et al., 2017a), and derive a new energy function and variational approximation to enable efficient learning and inference procedures. Additionally, this formulation allows us to consider the CRF kernels as latent variables and infer them from the data, together with the structured attention variables $\mathbf{m}^t$ and $\mathbf{v}^t$. We believe learning the kernels is important because it allows the CRF to weight the information flow depending on the content of the rather than keeping the same weights for all images.

We assume a generic CNN front-end providing a set of $S$ multi-scale feature maps $\mathbf{F} = \{\mathbf{f}_s\}_{s=1}^S$. To ease notation, we assume that each feature map has $P$ pixels and $C$ channels, but in practice these dimensions depend on the scale $s$. For each scale, we also consider the set of hidden variables $\mathbf{z}_s$ corresponding to $\mathbf{f}_s$, and $\mathbf{Z} = \{\mathbf{z}_s\}_{s=1}^S$. These hidden variables correspond to refined convolutional futures that incorporate information and attention from other feature maps, so as to better represent the key information for the pixel-level task at hand. Intuitively, the structured attention tensor should help refining the hidden variables to allow better performance at various pixel-level prediction tasks.

As in (Xu et al., 2017a), for every pair of *emitting* $e$ and *receiving* $r$ scales, we consider a dedicated attention tensor $\mathbf{a}_{e,r}$. Very importantly, in our case this attention tensor is structured following (1), and so we have a set of hidden spatial attention maps $\mathbf{M} = \{\mathbf{m}_{e,r}^t\}_{e,r,t=1}^{S,S,T}$ and hidden channel attention vectors $\mathbf{V} = \{\mathbf{v}_{e,r}^t\}_{e,r,t=1}^{S,S,T}$. More precisely, $\mathbf{m}_{e,r}^t \in \{0,1\}^P$ and $\mathbf{v}_{e,r}^t \in \{0,1\}^C$ are a binary spatial map and a stochastic channel-wise vector, hence $\sum_{c=1}^C v_{e,r}^{t,c} = 1$. In this way, we reduce ambiguity and ease the learning. This also means that the model is conceived to pay attention to only $T$ channels of the feature map. While this could seem limiting at first glance we remark that: (i) the model learns which are the *optimal* $T$ channels among the possible $C$ that have to be used to refine the hidden variables and (ii) the posterior distribution of $\mathbf{m}^t$ boils down to a convex combination of all channels, as it will appear clear when discussing the inference procedure.

### 2.1    ENERGY FUNCTION AND VARIATIONAL APPROXIMATION

Our model consists on three different latent variables: the hidden features $\mathbf{Z}$, and the hidden attention maps $\mathbf{M}$ and vectors $\mathbf{V}$. In addition, we also consider inferring the CRF kernels, denoted by $\mathbf{K}$ from

the data. More precisely, the energy function associated to the proposed models writes:

$$- E(\mathbf{Z}, \mathbf{M}, \mathbf{V}, \mathbf{K}, \mathbf{F}, \Theta) = \sum_s \sum_{p,c} \phi_z(z_r^{p,c}, f_r^{p,c})$$

$$+ \sum_{e,r} \sum_{p,c,p',c'} \sum_t m_{e,r}^{t,p} v_{e,r}^{t,c} \psi(z_r^{p,c}, z_e^{p',c'}, k_{r,p,c}^{e,p',c'}) + \phi_k(f_r^{p,c}, f_e^{p',c'}, k_{r,p,c}^{e,p',c'}), \qquad (2)$$

where $\phi_z$, $\phi_k$ and $\psi$ are potentials to be defined and $k_{r,p,c}^{e,p',c'}$ denotes the kernel value weighting the information flow from the $(p',c')$-th value of the feature map of scale $e$ to the $(p,c)$-th value of the feature map of scale $r$. Since the exact posterior distribution is not computationally tractable, we opt to approximate it with the following family of separable distributions:

$$p(\mathbf{Z}, \mathbf{M}, \mathbf{V}, \mathbf{K}|\mathbf{F}, \Theta) \approx q(\mathbf{Z}, \mathbf{M}, \mathbf{V}, \mathbf{K}) = q_z(\mathbf{Z}) q_m(\mathbf{M}) q_v(\mathbf{V}) q_k(\mathbf{K}). \qquad (3)$$

In that case, the optimal solution for each of the factors of the distribution is to take the expectation w.r.t. to all the others, for instance:

$$q_z(\mathbf{Z}) \propto \exp\left( - \mathbb{E}_{q_m(\mathbf{M})q_v(\mathbf{V})q_k(\mathbf{K})}\left\{ E(\mathbf{Z}, \mathbf{M}, \mathbf{V}, \mathbf{K}, \mathbf{F}, \Theta)\right\}\right). \qquad (4)$$

It can be shown that the optimal variational factors write:

$$q_z(z_r^{p,c}) \propto \exp\left( \phi_z(z_r^{p,c}, f_r^{p,c}) + \sum_{e \neq r} \sum_t \bar{m}_{e,r}^{t,p} \bar{v}_{e,r}^{t,c} \sum_{p',c'} \mathbb{E}_{q_z q_k}\{\psi(z_r^{p,c}, z_e^{p',c'}, k_{r,p,c}^{e,p',c'})\}\right),$$

$$q_m(m_{e,r}^{t,p}) \propto \exp\left( m_{e,r}^{t,p} \sum_c \bar{v}_{e,r}^{t,c} \sum_{p',c'} \mathbb{E}_{q_z,q_k}\{\psi(z_s^{p,c}, z_{s'}^{p',c'}, k_{r,p,c}^{e,p',c'})\}\right),$$

$$q_v(v_{e,r}^{t,c}) \propto \exp\left( v_{e,r}^{t,c} \sum_p \bar{m}_{e,r}^{t,p} \sum_{p',c'} \mathbb{E}_{q_z,q_k}\{\psi(z_s^{p,c}, z_{s'}^{p',c'}, k_{r,p,c}^{e,p',c'})\}\right), \qquad (5)$$

$$q_k(k_{r,p,c}^{e,p',c'}) \propto \exp\left( \phi_k(f_r^{p,c}, f_e^{p',c'}, k_{r,p,c}^{e,p',c'}) + \sum_t \bar{m}_{e,r}^{t,p} \bar{v}_{e,r}^{t,c} \mathbb{E}_{q_z}\{\psi(z_s^{p,c}, z_{s'}^{p',c'}, k_{r,p,c}^{e,p',c'})\}\right),$$

where $\bar{m}_{e,r}^{t,p} = \mathbb{E}_{q_m}\{m_{e,r}^{t,p}\}$ denotes the the posterior mean, and analogously for $\bar{v}_{e,r}^{t,c}$. This result also implies that thanks to the variational approximation in (3), the posterior distributions factorise in each of the variables above, e.g. $q_z(\mathbf{Z}) = \prod_{r,p,c=1}^{S,P,C} q_z(z_r^{p,c})$. The relation between the various hidden variables as for their inference is shown in Figure 2 (left). In addition, we also show the information flow between the hidden variables using arrows. Finally, on Figure 2 (right) we show the relation between the channel-wise and spatial attention variables and how the final structured attention tensor is computed.

## 2.2 INFERENCE WITH VISTA-NET

In order to construct an operative model we need to define the potentials $\phi_z$, $\phi_k$ and $\psi$. In our case, the unary potentials correspond to:

$$\phi_z(z_r^{p,c}, f_r^{p,c}) = -\frac{b_r^{p,c}}{2}(z_r^{p,c} - f_r^{p,c})^2, \quad \phi_k(f_r^{p,c}, f_e^{p',c'}, k_{r,p,c}^{e,p',c'}) = -\frac{1}{2}(k_{r,p,c}^{e,p',c'} - f_r^{p,c} f_e^{p',c'})^2, \quad (6)$$

where $b_s^{p,c} > 0$ is a weighting factor. $\psi$ is bilinear in the hidden feature maps:

$$\psi(z_r^{p,c}, z_e^{p',c'}, k_{r,p,c}^{e,p',c'}) = z_r^{p,c} k_{r,p,c}^{ep',c'} z_e^{p',c'}. \qquad (7)$$

Using the over bar notation also for the hidden features and kernels, e.g. $\bar{z}_s^{p,c} = \mathbb{E}_{q_z}\{z_s^{p,c}\}$, and by combining the kernel definitions (6) and (7) with the expression of the variational factors (5), we obtain the following update rules for the latent variables.

**Z-step.** It can be seen that the posterior distribution on $q_z$ is Gaussian with mean:

$$\bar{z}_s^{p,c} = \frac{1}{b_s^{p,c}}\left( b_s^{p,c} f_s^{p,c} + \sum_e \sum_t \bar{m}_{s,s'}^{t,p} \bar{v}_{s,s'}^{t,c} \sum_{p',c'} \bar{k}_{r,p,c}^{e,p',c'} \bar{z}_{s'}^{p',c'}\right) \qquad (8)$$

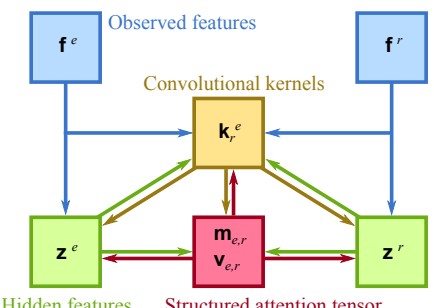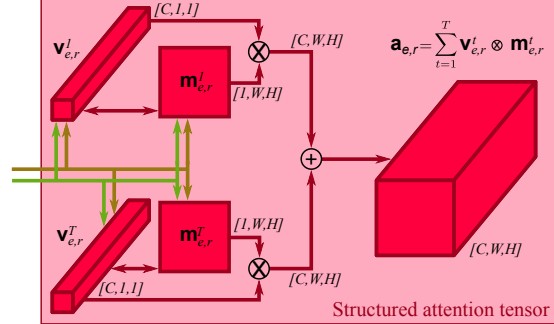

Figure 2: (left) Schematic representation of the various hidden variables in VISTA-Net. For each pair of emitting $e$ and receiving $r$ scales, their respective convolutional features $\mathbf{f}$ are shown in blue, their hidden variables $\mathbf{z}$ in green, the associated learned kernel $\mathbf{k}$ in yellow, and the channel-wise and spatial attention vector and matrix $\mathbf{v}$ and $\mathbf{m}$ in red. Arros of the corresponding color denote the flow of information when updating the variable. (right) The computational relationships between the channel-wise and spatial attention variables is shown, as well as the operations required to compute the final structured attention tensor $\mathbf{a}$.

This corresponds to the update rule obtained in (Xu et al., 2017a) with two remarkable differences. First, the posterior of the attention gate corresponds to the posterior of the structured tensor of rank $T$. Second, the impact of the neighboring features is weighted by the expected kernel value $\bar{k}_{r,p,c}^{e,p',c'}$.

**M-step.** The variational approximation leads to a Bernoulli distribution for $q_m(m_{e,r}^{t,p})$, which boils down to the following a posterior mean value using the sigmoid function $\sigma$:

$$\bar{m}_{e,r}^{t,p} = \sigma\Big( \sum_c \bar{v}_{e,r}^{t,c} \sum_{p',c'} \bar{z}_s^{p,c} \bar{k}_{r,p,c}^{e,p',c'} \bar{z}_{s'}^{p',c'} \Big). \tag{9}$$

**V-step.** It can be shown that the approximated posterior distribution is categorical, and that the expected value of each dimension of $\mathbf{v}_{e,r}^t$ can be computed using the softmax operator:

$$(\bar{v}_{e,r}^{t,c})_{c=1}^C = \text{softmax}\Big( \sum_p \bar{m}_{e,r}^{t,p} \sum_{p',c'} \bar{z}_s^{p,c} \bar{k}_{r,p,c}^{e,p',c'} \bar{z}_e^{p',c'} \Big)_{c=1}^C. \tag{10}$$

**K-step.** Finally, we need to derive the update rules for $\mathbf{K}$. By further deriving the corresponding variational posterior distribution, it can be shown that the a posterior distribution for the kernels is a Gaussian distribution with the following mean:

$$\bar{k}_{r,p,c}^{e,p',c'} = f_r^{p,c} f_e^{p',c'} + \sum_t \bar{m}_{e,r}^{t,p} \bar{v}_{e,r}^{t,c} \bar{z}_r^{p,c} \bar{z}_e^{p',c'}. \tag{11}$$

This solution is very straightforward, but since the kernels are estimated independently for each pair of receiving $(r, p, c)$ - emitting $(e, p', c')$ pixels, it has two major drawbacks. First, the kernel values are estimated without any spatial context. Second, given the large amount of kernel values, one must find a very efficient way to compute them. We propose to kill two birds with one stone by learning the kernels from the features using convolutional layers. By design, they take spatial context into account, and many popular libraries have efficient implementations of the convolution operation. The estimated kernel corresponding to the input channel $c'$ of scale $e$, $\mathbf{k}_r^{e,c'}$ is computed via a convolutional operation. The input of the convolution is a concatenation of the tensor $\mathbf{f}_r + \mathbf{z}_r \sum_{t=1}^T \mathbf{m}_{r,e}^t \otimes \mathbf{v}_{r,e}^t$ and the image $\mathbf{z}_e^{c'}$ resized to the spatial size of $\mathbf{f}_r$.

**Joint learning.** We implement the inference procedure described before within the neural network, on the top of the CNN front-end. Indeed, implementing all inference operations using available deep learning operators has two prominent advantages. First, we can perform the inference and learning the CNN front-end at the same time, within the same formalism and for the same aim. Second, this allows direct parallelisation of our method, speeding up training and inference.

The precise implementation goes as follows. Regarding $\bar{\mathbf{z}}_r$, we first apply message passing from the $e$-th scale to the $s_r$-th scale is performed with $\mathbf{z}_{e \to r} \leftarrow \bar{\mathbf{k}}_r^e \circledast \bar{\mathbf{z}}_e$, where $\circledast$ denotes the convolutional operation and $\bar{\mathbf{k}}_r^e$ denotes the corresponding learned convolution kernel. We then apply element-wise product with the corresponding structured attention tensor $\sum_{t=1}^{T} \bar{\mathbf{m}}_{e,r}^t \otimes \bar{\mathbf{v}}_{e,r}^t$. Finally we compute the element-wise sum with other emiting scales and the feature maps $\mathbf{f}_r$, see (8). Regarding $\bar{\mathbf{m}}_{e,r}$, we first compute the element-wise product between $\bar{\mathbf{z}}_r$ and $\mathbf{z}_{e \to r}$. The sum over channels weighted by $\bar{\mathbf{v}}_{e,r}$ is computed previous to applying pixel-wise sigmoid, see (9). Regarding $\bar{\mathbf{v}}_{e,r}$ we operate in a very similar fashion, but weighting each pixel with $\bar{\mathbf{m}}_{e,r}$ and then summing every channel independently, before applying softmax, see (10). Regarding $\bar{\mathbf{k}}_r^{e,c'}$, as discussed before, it is computed via a convolutional operation on the concatenations of $\mathbf{f}_{t_m} + \mathbf{g}_{t_m}$ and the image $\mathbf{z}_e^{c'}$ resized to the spatial size of $\mathbf{f}_r$. In terms of initialisation, we draw a random guess for $\mathbf{M}$ and $\mathbf{V}$, and set $\mathbf{Z}$ to $\mathbf{F}$. This allows us to update the kernels, then the other variables.

Once the hidden variables are updated, we use them to address several different pixel-wise prediction tasks involving continuous and discrete variables, including monocular depth estimation, surface normal estimation and semantic segmentation. Following previous works, the network optimization losses for these three tasks are a standard L2 loss (Xu et al., 2017c), a consine similarity loss (Eigen et al., 2014) and a cross-entropy loss (Chen et al., 2016a), respectively. The CNN front-end and VISTA-Net, are jointly trained end-to-end.

## 3 EXPERIMENTAL EVALUATION

### 3.1 DATASETS AND EXPERIMENTAL PROTOCOL

**Tasks and Datasets.** We demonstrate the effectiveness of VISTA-Net on two tasks: monocular depth estimation on the NYU-v2 (Silberman et al., 2012) and the KITTI (Geiger et al., 2013) datasets and semantic segmentation on the Pascal-Context (Mottaghi et al., 2014), the Pascal VOC2012 (Everingham et al., 2010) and the Cityscape (Cordts et al., 2016). We also conducted experiments on the surface normal estimation task on ScanNet (Dai et al., 2017) but due to lack of space the associated results are reported in the Appendix.

For NYU-v2 and KITTI we follow the experimental settings proposed by Eigen *et al.* (Eigen et al., 2014). For NYU-v2 we use 120K RGB-Depth pairs with a resolution of $480 \times 640$ pixels, acquired with a Microsoft Kinect device from 464 indoor scenes, using 249 scenes for training and 215 scenes (654 images) for test. For KITTI we specifically use 22,600 frames from 32 scenes for training and 697 frames from the rest 29 scenes for test.

For Pascal-Context we follow the works (Chen et al., 2016a; Zhang et al., 2018) and we consider the most frequent 59 classes. The remaining classes are masked during training and test. Pascal VOC2012 contains 20 classes divided in 10582 training, 1449 validation and 1456 test images. Our method is trained using the protocol described in (Zhong et al., 2020; Long et al., 2015). For Cityscape dataset, only the 5,000 finely annotated images are used in our experiments, split into 2,975/500/1,525 images for training, validation, and test.

**Evaluation Metrics.** To evaluate the performance on monocular depth estimation, we consider several metrics as in (Eigen & Fergus, 2015), including mean relative error (rel), root mean squared error (rms), mean log10 error (log10), and accuracy with threshold $t$ ($t \in \{1.25, 1.25^2, 1.25^3\}$). As for semantic segmentation, we consider two metrics following (Zhou et al., 2017; Zhang et al., 2018), *i.e.* pixel accuracy (pixAcc) and mean intersection over union (mIoU), averaged over classes.

**Implementation Details.** VISTA-Net is implemented in *Pytorch*. The experiments are conducted on four Nvidia Quadro RTX 6000 GPUs, each with 24 GB memory. The ResNet-101 architecture pretrained on ImageNet (Deng et al., 2009) is considered in all the experiments for initializing the backbone network of VISTA-Net except in for the experiments on the Cityscape dataset where we choose HRNet V2-W48 (whose complexity is comparable to dilated-ResNet-101) for fair comparison with previous works. Our model can be used for effective deep feature learning in both single-scale and multi-scale contexts. To boost the performance, following previous works (Xie & Tu, 2015; Xu et al., 2017a), we also consider features output by different convolutional blocks of a CNN backbone (*e.g.* res3c, ref4f, ref5d of a ResNet-50). For the semantic segmentation task, we use a learning rate of 0.001 on Pascal-context and Pascal-VOC 2012 and 0.01 on cityscapes with a momentum of 0.9 and a weight decay of 0.0001 using a polynomial learning rate scheduler as pre-

Table 1: Depth Estimation: KITTI dataset. Only monocular estimation methods are reported.

| Method | Error (lower is better) | | | | Accuracy (higher is better) | | |
|---|---|---|---|---|---|---|---|
| | abs-rel | sq-rel | rms | log-rms | $\delta<1.25$ | $\delta<1.25^2$ | $\delta<1.25^3$ |
| CC (Ranjan et al., 2019) | 0.140 | 1.070 | 5.326 | 0.217 | 0.826 | 0.941 | 0.975 |
| Bian et al. (2019) | 0.137 | 1.089 | 5.439 | 0.217 | 0.830 | 0.942 | 0.975 |
| $S^3$Net(Cheng et al., 2020a) | 0.124 | 0.826 | 4.981 | 0.200 | 0.846 | 0.955 | 0.982 |
| MS-CRF (Xu et al., 2017c) | 0.125 | 0.899 | 4.685 | - | 0.816 | 0.951 | 0.983 |
| AG-CRF (Xu et al., 2017a) | 0.126 | 0.901 | 4.689 | 0.157 | 0.813 | 0.950 | 0.982 |
| Monodepth2 (Godard et al., 2019) | 0.115 | 0.903 | 4.863 | 0.193 | 0.877 | 0.959 | 0.981 |
| pRGBD(Tiwari et al., 2020) | 0.113 | 0.793 | 4.655 | 0.188 | 0.874 | 0.96 | 0.983 |
| SGDepth(Klingner et al., 2020) | 0.107 | 0.768 | 4.468 | 0.180 | 0.891 | 0.963 | 0.982 |
| Johnston & Carneiro (2020) | 0.106 | 0.861 | 4.699 | 0.185 | 0.889 | 0.962 | 0.982 |
| Shu et al. (2020) | 0.104 | 0.729 | 4.481 | 0.179 | 0.893 | 0.965 | 0.984 |
| DORN (Fu et al., 2018) | 0.072 | 0.307 | 2.727 | 0.120 | 0.932 | 0.984 | 0.994 |
| Yin et al. (2019) | 0.072 | - | 3.258 | 0.117 | 0.938 | 0.990 | 0.998 |
| PackNet-SfM (Guizilini et al., 2020) | 0.071 | 0.359 | 3.153 | 0.109 | 0.944 | 0.990 | 0.997 |
| Lee et al. (2019) | **0.061** | 0.261 | 2.834 | 0.099 | 0.954 | 0.992 | 0.998 |
| VISTA-Net (ours) | 0.063 | **0.255** | **2.776** | **0.099** | 0.954 | **0.993** | **0.998** |

| Method | Error (lower is better) | | | Accuracy (higher is better) | | | |
|---|---|---|---|---|---|---|---|
| | rel | log10 | rms | $\delta<1.25$ | $\delta<1.25^2$ | $\delta<1.25^3$ | |
| PAD-Net (Xu et al., 2018) | 0.214 | 0.091 | 0.792 | 0.643 | 0.902 | 0.977 | |
| Li et al. (2017) | 0.152 | 0.064 | 0.611 | 0.789 | 0.955 | 0.988 | |
| CLIFFNet(Lijun et al., 2020) | 0.128 | 0.171 | 0.493 | 0.844 | 0.964 | 0.991 | |
| Laina et al. (2016) | 0.127 | 0.055 | 0.573 | 0.811 | 0.953 | 0.988 | |
| MS-CRF (Xu et al., 2017c) | 0.121 | 0.052 | 0.586 | 0.811 | 0.954 | 0.987 | |
| Lee & Kim (2020) | 0.119 | 0.5 | - | 0.87 | 0.974 | 0.993 | |
| AG-CRF (Xu et al., 2017a) | 0.112 | 0.051 | 0.526 | 0.818 | 0.960 | 0.989 | |
| DORN (Fu et al., 2018) | 0.115 | 0.051 | 0.509 | 0.828 | 0.965 | 0.992 | |
| Xia et al. (2020) | 0.116 | - | 0.512 | 0.861 | 0.969 | 0.991 | |
| Yin et al. (2019) | **0.108** | **0.048** | 0.416 | 0.875 | 0.976 | 0.994 | |
| Lee et al. (2019) | 0.113 | 0.049 | 0.407 | 0.871 | 0.977 | 0.995 | |
| VISTA-Net (ours) | 0.111 | **0.048** | **0.393** | **0.881** | **0.979** | **0.996** | |

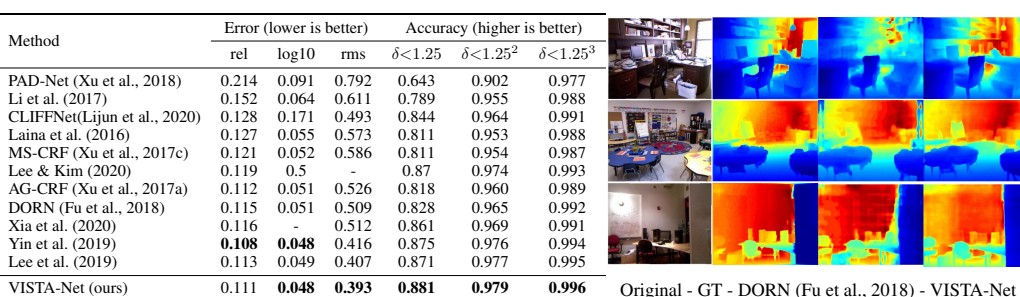

Original - GT - DORN (Fu et al., 2018) - VISTA-Net

Figure 3: Depth estimation: quantitative (left) and qualitative (right) comparison on NYU-v2.

viously done in (Zhang et al., 2018; Chen et al., 2016a). For the monocular depth estimation task, the learning rate is set to $10^{-4}$ with weight decay of 0.01. The Adam optimizer is used in all our experiments with a batch size of 8 for monocular depth estimation and 16 for semantic segmentation. The total training epochs are set to 50 for depth prediction experiments, to 150 for Pascal-context and Pascal VOC 2012 datasets and to 500 for the Cityscapes dataset.

## 3.2 EXPERIMENTAL RESULTS AND ANALYSIS

**Monocular Depth Estimation.** Comparative results on KITTI dataset are shown in Table 1. We propose a comparison with state of the art models such as (Eigen et al., 2014; Ranjan et al., 2019; Bian et al., 2019; Godard et al., 2019; Fu et al., 2018; Yin et al., 2019; Lee et al., 2019; Guizilini et al., 2020). In addition we demonstrate the effectiveness of our VISTA-Net comparing with MS-CRF (Xu et al., 2017c), a previous approach which exploit a probabilistic framework for multi-scale feature learning but does not consider an attention mechanisms. Our approach is superior, thus demonstrating the effectiveness of the proposed attention model. We also compare with AG-CRF (Xu et al., 2017a) adapting their model to the monocular depth estimation problem. Also in this case VISTA-Net outperforms the competitor confirming the importance of having a joint structured spatial- and channel-wise attention model. Note that AG-CRF (Xu et al., 2017a) and VISTA-Net are compared using the same backbone. In order to demonstrate the competitiveness of our approach in an indoor scenario we also report the results on NYUD-V2 dataset in Fig. 3. Similarly to the experiments on KITTI, VISTA-Net outperforms both state of the art approaches and previous methods based on attention gates and CRFs (Xu et al., 2017c;a).

**Semantic Segmentation.** We first compare VISTA-Net with the most recent methods on the Pascal-Context dataset, including (Zhang et al., 2018; Fu et al., 2019; Zhu et al., 2019; Ding et al., 2019; Zhang et al., 2019b; Wang et al., 2020; He et al., 2019). As for the depth estimation task, also in this case we evaluate the performance of AG-CRF Xu et al. (2017a), adapting the original code to the semantic segmentation task. VISTA-Net, as shown in Table 2, is 0.6 points better according to the mIoU metric than the best available method, *i.e.* AG-CRF. Importantly, VISTA-Net outperforms EncNet (Zhang et al., 2018), which uses only channel-wise attention, as well as DANet (Fu et al.,

Table 2: Semantic Segmentation on PASCAL-Context. D-ResNet-101 denotes Dilated ResNet-101.

| Method | Backbone | pixAcc% | mIoU% |
|---|---|---|---|
| CFM (VGG+MCG) (Dai et al., 2015b) | VGG-16 | - | 34.4 |
| DeepLab-v2 (Chen et al., 2016a) | VGG-16 | - | 37.6 |
| FCN-8s (Long et al., 2015) | VGG-16 | 50.7 | 37.8 |
| BoxSup (Dai et al., 2015a) | VGG-16 | - | 40.5 |
| ConvPP-8s (Xie et al., 2016) | VGG-16 | - | 41.0 |
| PixelNet (Bansal et al., 2017) | VGG-16 | 51.5 | 41.4 |
| HRNetV2 (Wang et al., 2020) | - | - | 54.0 |
| EncNet (Zhang et al., 2018) | D-ResNet-101 | 79.23 | 51.7 |
| DANet (Fu et al., 2019) | D-ResNet-101 | - | 52.6 |
| ANN (Zhu et al., 2019) | D-ResNet-101 | - | 52.8 |
| SpyGR (Li et al., 2020a) | ResNet-101 | - | 52.8 |
| SANet (Zhong et al., 2020) | ResNet-101 | 80.6 | 53.0 |
| SVCNet (Ding et al., 2019) | ResNet-101 | - | 53.2 |
| CFNet (Zhang et al., 2019b) | ResNet-101 | - | 54.0 |
| APCNet (He et al., 2019) | D-ResNet-101 | - | 54.7 |
| AG-CRF (Xu et al., 2017a) | D-ResNet-101 | 80.8 | 54.8 |
| VISTA-Net (ours) | D-ResNet-101 | **81.1** | **55.4** |

Table 3: Semantic Segmentation: results on both Cityscapes validation and testing set (trained on the standard training set) (left) and on PASCAL VOC 2012 validation set (right). D-ResNet-101 means Dilated-ResNet-101. (*) indicates COCO pretrained weights.

| Method | Backbone | Test set | mIoU |
|---|---|---|---|
| Dynamic (Li et al., 2020b) | Layer33-PSP | Val | 79.7 |
| SpyGR (Li et al., 2020a) | ResNet-101 | Val | 80.5 |
| CCNet (Huang et al., 2019b) | ResNet-101 | Val | 81.3 |
| Panoptic-DeepLab (Cheng et al., 2020b) | D-ResNet-101 | Val | 81.5 |
| CDGCNet (Hu et al., 2020) | D-ResNet-101 | Val | 81.9 |
| VISTA-Net | HRNetV2-W48 | Val | **82.3** |
| PSANet (Zhao et al., 2018) | D-ResNet-101 | Test | 78.6 |
| PAN (Li et al., 2018) | D-ResNet-101 | Test | 78.6 |
| AAF Ke et al. (2018) | D-ResNet-101 | Test | 79.1 |
| HRNet (Wang et al., 2020) | HRNetV2-W48 | Test | 80.4 |
| Dynamic (Li et al., 2020b) | Layer33-PSP | Test | 80.7 |
| VISTA-Net | HRNetV2-W48 | Test | **81.4** |

| Method | Backbone | mIoU% |
|---|---|---|
| DeepLabV3 (Chen et al., 2017) | D-ResNet-101 | 75.7 |
| Dynamic (Li et al., 2020b) | Layer33 | 79.0 |
| Res2Net (Gao et al., 2019) | Res2Net-101 | 80.2 |
| DANet (Fu et al., 2019) | ResNet-101 | 80.4 |
| Auto-Deeplab (Liu et al., 2019)* | ResNet-101 | 82.0 |
| EncNet (Zhang et al., 2018) | D-ResNet-101 | 85.9 |
| SANet (Zhong et al., 2020)* | ResNet-101 | 86.1 |
| VISTA-Net | D-ResNet-101 | **89.8** |

2019) and SANet(Zhong et al., 2020), which considers inter-dependencies both at spatial and channel level in their attention model. In Table 3(right) are shown results on PASCAL VOC2012. Again, our method outperforms EncNet (Zhang et al., 2018), SANet (Zhong et al., 2020) and DANet (Fu et al., 2019). In particular, VISTA-Net is 3.7 points better according to the mIoU metric than the best available method, *i.e.* SANet. Finally, Table. 3(left) reports the results on Cityscape. As in the previous two datasets VISTA-Net outperforms the competitors (by nearly one point mIoU). Additional results are reported in the Appendix.

**Ablation Study.** We also performed an ablation study on the Pascal-context dataset to further demonstrate each proposed component's impact. Fig. 5 (left) shows that the performance of VISTA-Net degrades not only when the model does not employ the structured attention mechanism, but also when only channel-wise or spatial-wise attention is used. Moreover, we can also see the advantage of using the proposed probabilistic formulation for joint modeling both spatial- and channel-wise attention in a principled manner. Interestingly, the performance achieved in each of the variants (spatial, channel, no probabilistic formulation) is similar. This leads us to believe that the proposed method's competitive advantage is in combining structured attention with a probabilistic formulation. Notably, the feature refinement through message passing seems to be the most crucial contribution to improve the performance. For the sake of completeness, we also report the results of DANet, and of AG-CRF (which corresponds to the Multiple-scale/Spatial setting in). Finally, in Fig.5 we show the performance of VISTA-Net for different values of the tensor rank $T$. It is important to notice that the framework reaches better performance when $T$ is higher. Fig. 4 clearly illustrate the perceptual improvement in segmentation masks obtained with higher values of the attention tensor rank. Additional examples are provided in the supplementary material.

## 4 CONCLUSIONS

In this paper we proposed a novel approach to improve the learning of deep features representations for dense pixel-wise prediction tasks. Our approach seamlessly integrates a novel structured atten-

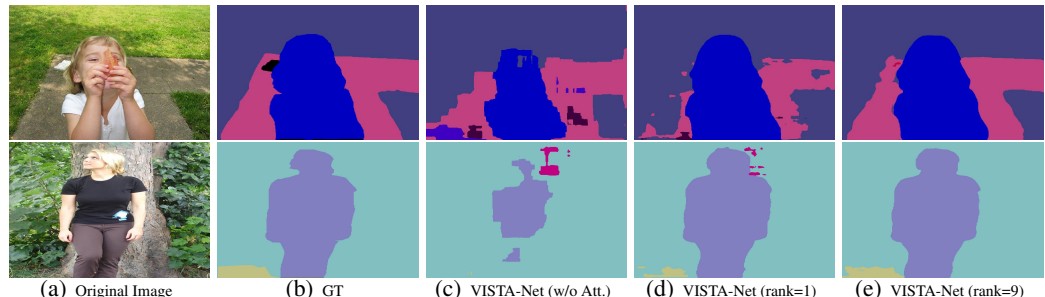

|                      |                  |                       |                       |
|:--------------------:|:----------------:|:---------------------:|:---------------------:|
| (a) Original Image | (b) GT | (c) VISTA-Net (w/o Att.) | (d) VISTA-Net (rank=1) | (e) VISTA-Net (rank=9) |

Figure 4: Semantic segmentation maps obtained with VISTA-Net on the Pascal-Context dataset.

| Scales | Structured Attention | Probabilistic | mIoU | PixAcc |
|---|---|---|---|---|
| DANet (Fu et al., 2019) | Separate Attention | No | 52.6 | - |
| Single scale | No structure | Yes | 51.7 | 78.9 |
| | Spatial | Yes | 53.0 | 79.7 |
| | Channel | Yes | 53.1 | 79.8 |
| | Low-rank tensor | No | 53.2 | 79.9 |
| | Low-rank tensor | Yes | 53.9 | 80.3 |
| Multiple scale | No structure | Yes | 52.8 | 79.5 |
| | Spatial | Yes | 54.8 | 80.8 |
| | Channel | Yes | 54.6 | 80.6 |
| | Low-rank tensor | No | 54.7 | 80.7 |
| | Low-rank tensor | Yes | **55.4** | **81.1** |

Figure 5: Ablation study on the Pascal-context dataset: performance of VISTA-Net (left) for different attention mechanisms and scales, (right) for different values of the tensor rank $T$.

tion model within a probabilistic framework. In particular, we proposed to structure the attention tensors as the sum of $T$ rank-one tensors, each being the tensor-product of a spatial attention map and a channel attention vector. These two kinds of variables are jointly learned within the probabilistic formulation made tractable thanks to the variational approximation. The proposed structured attention is rich enough to capture complex spatial- and channel-level inter-dependencies, while being efficient to compute. The overall optimisation of the probabilistic model and of the CNN front-end is performed jointly. Extensive experimental evaluations show that VISTA-Net outperforms state-of-the-art methods on several datasets, thus confirming the importance of structuring the attention variables for dense pixel-level prediction tasks.

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

## A    APPENDIX: ADDITIONAL EXPERIMENTAL RESULTS

In this section we report additional quantitative and qualitative results.

**Cityscapes.** Table 4 reports the results of VISTA-Net trained on the training+validation set and tested on the test set of Cityscapes dataset. We focus on comparing our method with recent attention networks, including (Zhang et al., 2019a; Fu et al., 2019; Zhang et al., 2020; Huang et al., 2019b; Choi et al., 2020). This table confirms that our structured attention module is better than the other attention methods and sets the new state-of-the-art on this dataset.

**Surface normal Estimation.** We conduct experiments on the ScanNet dataset. ScanNet is a large scale RGB-D dataset for 3D scene understanding, we follow the protocol proposed in (Dai et al., 2017) with a split of 189,916/20,942 images for training and test respectively. The surface normal prediction performance is evaluated using five metrics. We compute the per-pixel angle distance between prediction and ground-truth, mean and median for valid pixels with given ground-truth normal. In addition to mean and median, we also compute the fraction of pixels with angle difference w.r.t. ground-truth less than $t \in [11.25°, 22.5°, 30°]$ as used in (Eigen et al., 2014). We compare VISTA-Net with other state-of-the-art RGB-based methods including (Bansal et al., 2016; Zhang et al., 2017; Qi et al., 2018; Huang et al., 2019a). Quantitative results are shown in Table. 5. Our

Table 4: Semantic segmentation results on Cityscapes test set (learned on the train+val set, multi-scale and flipping). D-ResNet-101 is short for Dilated-ResNet-101.

| Method | Backbone | mIoU | iIoU cla | IoU cat. | iIoU cat. |
|---|---|---|---|---|---|
| DeepLab (Chen et al., 2016a) | D-ResNet-101 | 70.4 | 42.6 | 86.4 | 67.7 |
| PADNet (Xu et al., 2018) | D-ResNet-101 | 80.3 | 58.8 | 90.8 | 78.5 |
| Dynamic (Li et al., 2020b) | Layer33-PSP | 80.7 | - | - | - |
| SVCNet (Ding et al., 2019) | ResNet-101 | 81.0 | - | - | - |
| ANN (Zhu et al., 2019) | D-ResNet-101 | 81.3 | - | - | - |
| CCNet (Huang et al., 2019b) | D-ResNet-101 | 81.4 | - | - | - |
| DANet (Fu et al., 2019) | D-ResNet-101 | 81.5 | - | - | - |
| DGMN (Zhang et al., 2020) | D-ResNet-101 | 81.6 | - | - | - |
| SpyGR (Li et al., 2020a) | ResNet-101 | 81.6 | - | - | - |
| HRNet (Wang et al., 2020) | HRNetV2-W48 | 81.6 | 61.8 | 92.1 | 82.2 |
| ACFNet (Zhang et al., 2019a) | ResNet-101 | 81.8 | - | - | - |
| DGCNet (Zhang et al., 2019c) | ResNet-101 | 82.0 | - | - | - |
| HANet (Choi et al., 2020) | ResNext-101 | 82.1 | - | - | - |
| VISTA | HRNetV2-W48 | **82.2** | 62.7 | 91.9 | 82.1 |

method outperforms the FrameNet by more than 3 % in $11.25°$ and leaves behind by a significant margin the other methods. In addition we show some qualitative examples in Fig. 6. These results clearly indicate that predictions are extremely accurate also in the case of objects (i.e. waste bin, water closet, chair supports, etc.).

**Depth estimation: qualitative results.** In Fig. 7 is shown a qualitative comparison of our method with DORN (Fu et al., 2018). Results indicate that VISTA-Netgenerates better depth maps, in particular one can appreciate the opening of the sky and the smoothness of the prediction on the sides. Fig 8 shows a similar comparison done on NYU-D dataset. The same accuracy in the prediction is visible also in this case, objects are more distinguishable w.r.t. DORN (e.g. the bathtub in row 2 and the desks in row 5). Finally, we also provided the computed attention maps on some sample images in KITTI dataset in Fig. 13. As expected, the final structured attention tensors manage to capture important information among different depths. For example, in the fourth row, structured attention focus on farthest frost, middle jungle, and close road.

**Semantic segmentation: qualitative results.** In Fig. 9 we propose a few qualitative results on Pascal-context dataset. In the figure is shown the importance of the attention model and the result obtained increasing the iterations. In the odd rows are shown the misclassified pixels (in black). The image shows clearly how the proposed iterative approach based on message passing is beneficial for the final prediction. Additional qualitative results on Cityscapes dataset are shown in Fig. 10. Also in this case, the segmentation maps produced by VISTA-Netare more precise w.r.t. those of the competitors (Fu et al., 2019; Wang et al., 2020), in particular one can notice the superior accuracy in the right hand side of the images where VISTA-Netshows better visual predictions.

**Semantic segmentation: computational cost.** In Table. 6 we show the results of our experiments in order to analyze the computational cost of our method. In particular, we perform an analysis on the Pascal-context dataset and at varying $T$. In the table FPS means Frames Per Second. As expected, when the rank increases we also observe an increase in term of parameters and a reduction in term of speed.

**Additional qualitative maps.** Fig. 11 shows different visualizations regarding the learned structured attention on an image from the Pascal-Context dataset. The first row shows the original image, together with four slices (channels) of the overall structured attention tensor $\mathbf{a}$ as defined in (1). The second row shows the $T = 5$ spatial maps of the structured tensor $\mathbf{m}^t$. While the latter seem to be spread all along the dog's body with different shapes, we observe that by optimally combining the $\mathbf{m}^t$'s using the $\mathbf{v}^t$'s, different slices of the final structured attention tensor are able to on different important parts of the dog: the head, the body, the tail and the rear paws, thus allowing to take much more accurate pixel-level predictions for segmentation.

Meanwhile, Fig. 12 depicts segmentation maps obtained on the Pascal-Context dataset using different versions of our method. In particular, we visualize (c) VISTA-Net w/o Attention, (d) VISTA-Net w/o Spatial Attention, (e) VISTA-Net w/o Channel Attention and (f) VISTA-Net (full model). From left to right, the results become more similar to the ground truth indicating the clear advantage of our proposed attention model.

Table 5: Quantitative evaluation on surface normal prediction.

| Methods | Error metric | | Accuracy metric | | |
|---|---|---|---|---|---|
| | mean | median | 11.25 | 22.5 | 30 |
| Skip-Net(Bansal et al., 2016) | 26.2 | 20.6 | 28.8 | 54.3 | 67.0 |
| Zhang*et al.* (Zhang et al., 2017) | 23.3 | 16.0 | 40.4 | 63.1 | 71.9 |
| GeoNet (Qi et al., 2018) | 19.8 | 11.3 | 49.7 | 70.4 | 77.7 |
| FrameNet (Huang et al., 2019a) | 15.3 | 8.1 | 60.6 | 78.6 | 84.7 |
| VISTA-Net | **15.1** | **7.5** | **63.8** | **80.0** | **85.2** |

Table 6: Pascal-context dataset: computational cost analysis for different values of $T$.

| rank | IoU | pixacc | parameters | FPS |
|---|---|---|---|---|
| 0 | 54.2 | 80.4 | 45.80M | 1.106 |
| 1 | 54.6 | 80.7 | 49.85M | 1.075 |
| 3 | 54.8 | 80.8 | 52.68M | 1.011 |
| 5 | 54.5 | 80.4 | 54.89M | 1.068 |
| 7 | 55.3 | 81.1 | 56.79M | 0.957 |
| 9 | 55.4 | 81.1 | 58.85M | 0.868 |

**Network detail.** The overview of VISTA-Net is depicted in Fig. 14 (a). From left to right features are extracted from the backbone after layers 2 to 4 and then are fed to the structured attention gate module (a detailed view is shown in Fig. 14 (b)) producing the attention maps. For each emitting and receiving scale (map), one structured attention gate is computed and exploited to gate the information sent from the emitting feature map to the receiving feature map. An inner view of the attention gate and its connections are shown in Fig. 14 (b). The blue cross $\otimes$ denotes the convolution operation while green cross refers to the conditional kernel prediction. The symbols $\odot$ and $\oplus$ denote element-wise multiplication and addition, respectively while $\sigma$ ,$\copyright$ refer to the sigmoid function and feature concatenation. Finally the color code for arrows is green for M-step, yellow for V-step and red for K-step. The algorithm is also described in Algorithm 1.

---

**Algorithm 1:** Our structured attention algorithm (VISTA-Net)

**Input :**
- $f_e$ – emitting feature, size is $[B, C, H, W]$
- $f_r$ – receiving feature, size is $[B, C, H, W]$

**Output:**
- $\hat{f}_r$ – updated receiving feature, size is $[B, C, H, W]$

1   $f_{concat} \leftarrow concat(f_e, f_r)$
2   $L \leftarrow Conv2d(f_{concat})$
3   $L_{se \rightarrow sr} \leftarrow Conv2d(f_e)$
4   $L_{sr \rightarrow se} \leftarrow Conv2d(f_r)$
5   $h_{se} \leftarrow unfold(f_e)$
6   $h_{sr} \leftarrow unfold(f_r)$
7   $A \leftarrow \sigma(L \cdot f_e + L_{se \rightarrow sr} \cdot h_{se} + L_{sr \rightarrow se} \cdot h_{sr})$
8   **for** $t \leftarrow 1$ **to** $T$ **do**
9      $A_{ch} \leftarrow randn(B, C, 1, 1)$
10     $\bar{A}_{ch} \leftarrow softmax(A_{ch})$
11     $A_{sp} \leftarrow \sum_c^C (\bar{A}_{ch} \cdot A)$
12     $\bar{A}_{sp} \leftarrow sigmoid(A_{sp})$
13     $A_{ch} \leftarrow \sum_i^H \sum_j^W (\bar{A}_{sp} \cdot A)$
14     $\bar{A}_{ch} \leftarrow softmax(A_{ch})$
15   **end for**
16   $\hat{f}_r \leftarrow (L \cdot h_{se}) \cdot (\bar{A}_{ch} \cdot \bar{A}_{sp} \cdot A) + f_r$
17   **return** $\hat{f}_r$

---

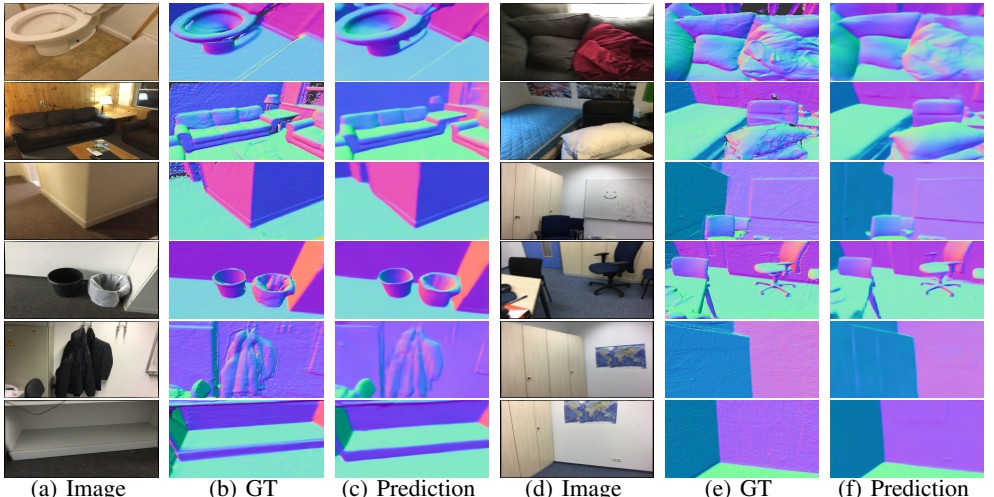

| (a) Image | (b) GT | (c) Prediction | (d) Image | (e) GT | (f) Prediction |

Figure 6: ScanNet dataset. Qualitative examples on the surface normal prediction task.

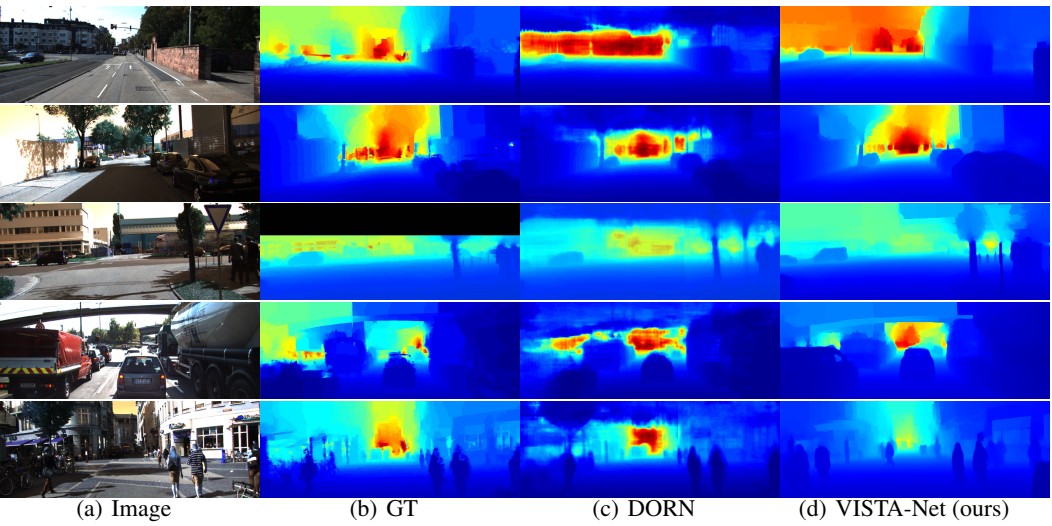

| (a) Image | (b) GT | (c) DORN | (d) VISTA-Net (ours) |

Figure 7: KITTI dataset. Qualitative examples on the monocular depth prediction task. We performed bilinear interpolation on the sparse ground-truth depth maps for better visualization.

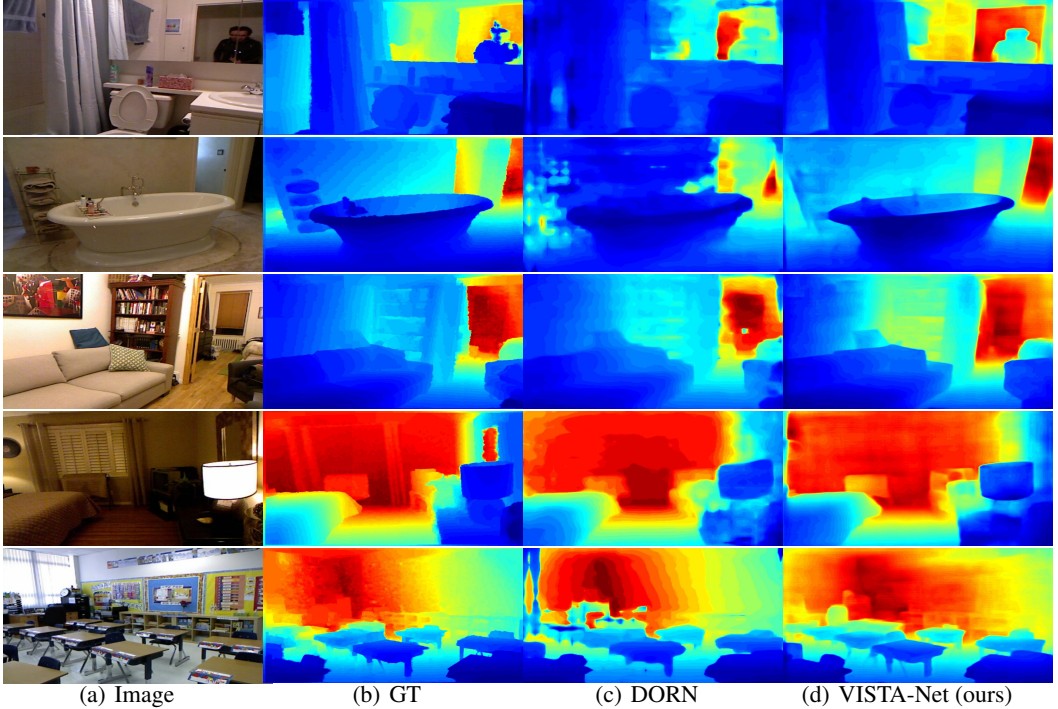

|  (a) Image | (b) GT | (c) DORN | (d) VISTA-Net (ours) |

Figure 8: NYU-V2 dataset. Qualitative examples on the monocular depth prediction task. We performed bilinear interpolation on the sparse ground-truth depth maps for better visualization.

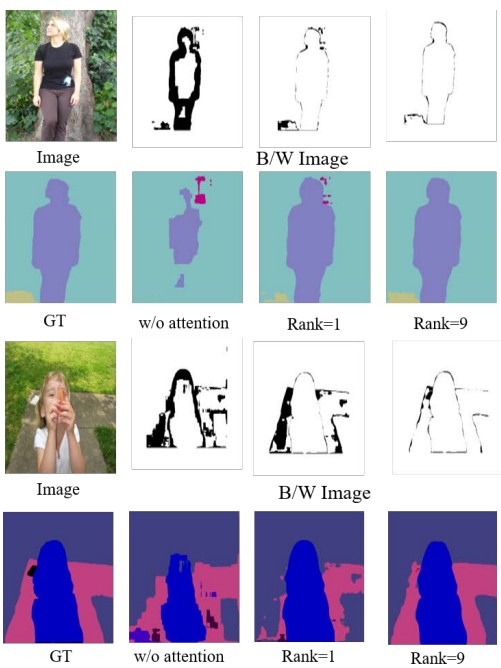

Figure 9: Pascal-context dataset. Comparison of different variations of VISTA-Net.

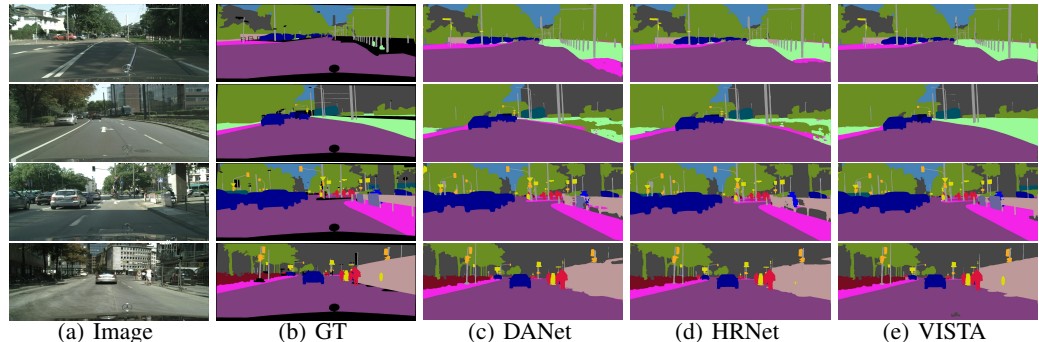

(a) Image          (b) GT          (c) DANet          (d) HRNet          (e) VISTA

Figure 10: Qualitative semantic segmentation results on Cityscapes dataset.

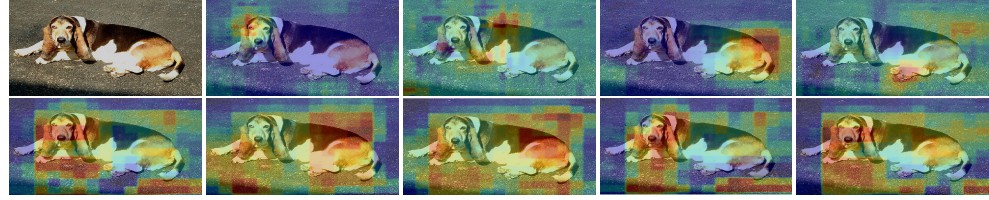

Figure 11: First row: original image and four channels (slices) of the structured attention tensor $\mathbf{a}$, see (1). Second row, the $T = 5$ attention maps $\mathbf{m}^t$, $t = 1, \ldots, 5$.

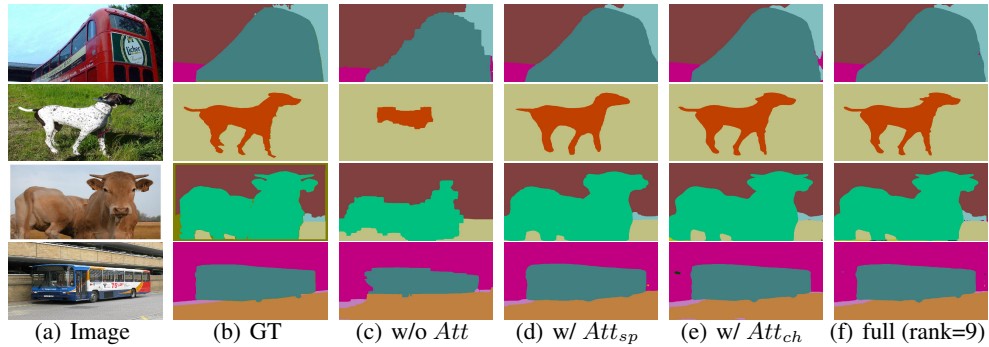

(a) Image      (b) GT      (c) w/o $Att$      (d) w/ $Att_{sp}$      (e) w/ $Att_{ch}$      (f) full (rank=9)

Figure 12: Pascal-context dataset. Segmentation maps using different attention models in VISTA-Net.

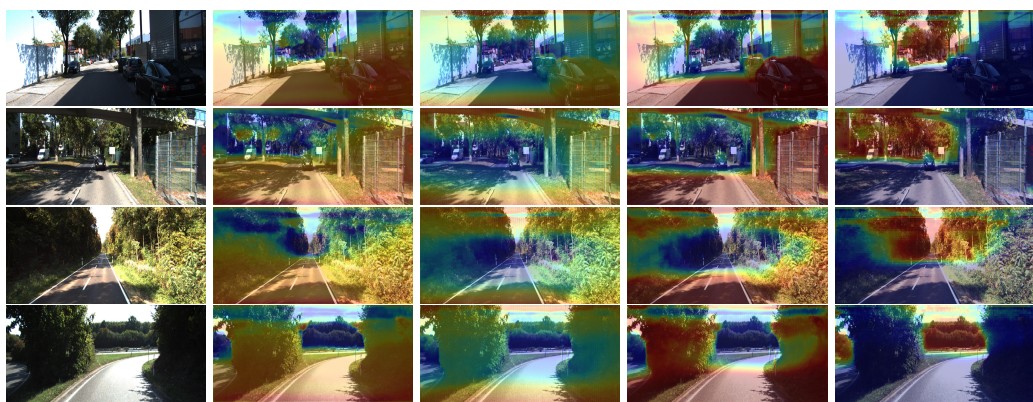

Figure 13: Qualitative structured attention examples of monocular depth prediction on the KITTI raw dataset. First column is original image and next four columns is structured attention.

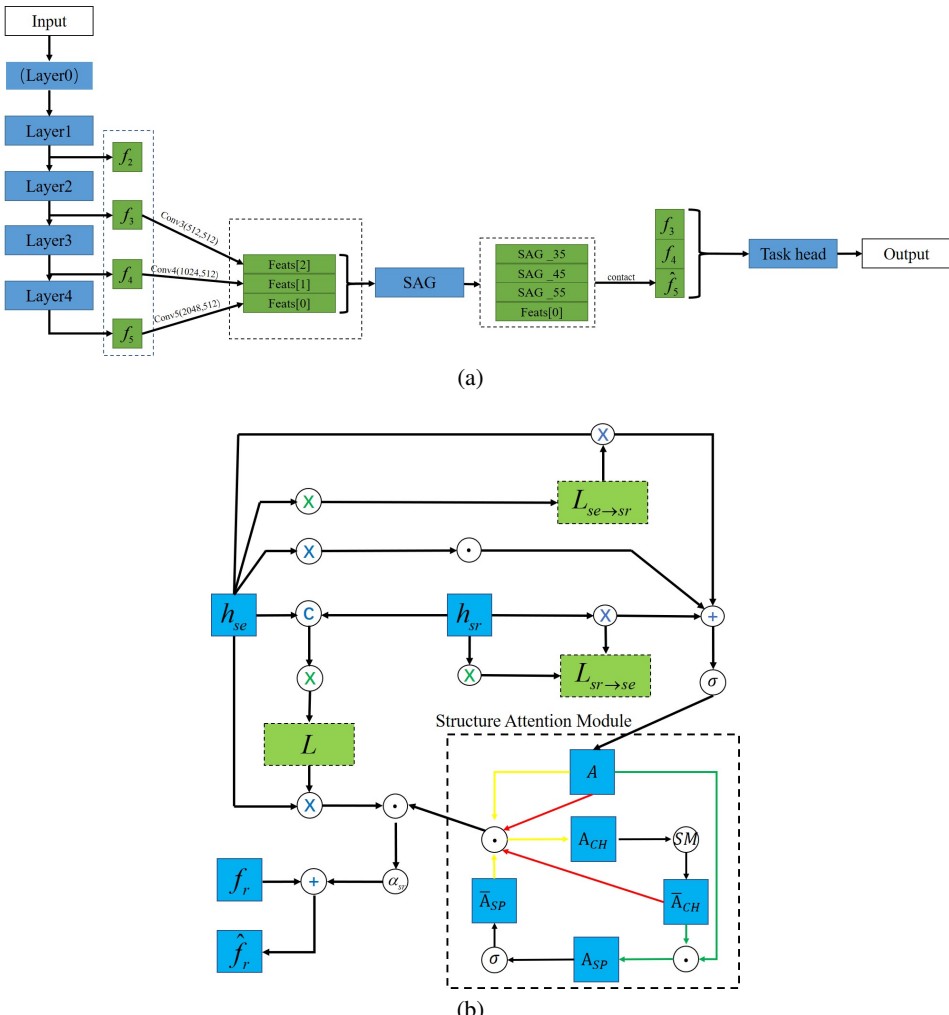

Figure 14: The overview of VISTA-Net (a). SAG is short for structure attention gate module. In (b) is shown a detailed view of the attention-gate model

