# OpenReview forum: "Variational Structured Attention Networks for Dense Pixel-Wise Prediction"
_ICLR.cc/2021/Conference — Reject_

### Official Review · AnonReviewer2 · 2020-10-22
**There some unclear statements in this paper. However, generally speaking, this paper proposes a novel structured attention mechanism for the pixel-wise prediction tasks. Extensive experiments have been conducted to demonstrate the effectiveness and superiority of the proposed method. Therefore, I think this paper may be considered to accept.**

**Rating:** 6
**Confidence:** 4

**Review:**

This paper proposes a novel structure attention mechanism with a probabilistic CRF-like inference framework for the pixel-wise prediction, which not only models the spatial-wise dependencies but also considers the channel-level dependencies using an attention tensor. This attention mechanism is obviously different from the existing attention mechanism. Additionally, many quantitative and qualitative experimental results on three pixel-wise prediction tasks have demonstrated the superiority of the proposed method. The proposed attention mechanism is useful and flexible, which could be integrated into any pixel-wise prediction frameworks in theory. However, I think there exist some disadvantages:

1. There are some unclear statements in the paper. For instance, the authors say “the attention tensor is nothing but the sum of T rank-1 tensors”, but how are these rank-1 tensors generated by the spatial attention map and the channel-wise attention vector? Is each rank-1 tensor associated with the special regions of different objects? If the authors visualize the attention map, it would be better to understand. Moreover, the overall architecture of the proposed method is not clear. The authors do not specify the details of the multi-scale features, including the dimensions of them, where they come from, etc. The dimensions of the most important variables are also not specified, which makes the paper is hard to follow.

2. In the ablation study, when the model is ‘no structure’ on a single scale, does it denote the model does not integrate any attention mechanism? If so, when only considering the spatial or channel attention, why does it outperform DANet that includes dual attentions? If not, I think the comparison is not fair.

3. Although the performance of the proposed attention mechanism outperforms most existing methods, the authors do not report the runtimes and parameters. Moreover, whether does it significantly burden the complexity of the model with the increase of T?

4. There are some grammar mistakes and typos. For example, “Since the exact a posteriori distribution is not computationally tractable, ...”, “Again, our method not outperforms …”

---

> ### Author Response · Authors · 2020-11-14
> **Response to Reviewer 4**
>
> We clarified how the channel-wise and spatial attention variables relate to the structured attention tensor in the new Figure 2 and accompanying text. The spatial attention maps are learned, and are supposed to aggregate regions that behave similarly in terms of attention. We already provided attention maps for the Pascal-context dataset (Fig.10) and we have now added additional qualitative results in Fig. 12. Implementation details about multi-scale features are provided in the appendix (Network details). Answering reviewer's specific question, we consider the output of layer 2 to 4 of Resnet-101 that are fed to the structured attention task. Their dimensions are 512,1024 and 2048 respectively for all employed dataset with the exception of cityscape. In cityscape dataset, we use 2nd to 4th output of HRNet-V2 stage II where dimensions are 96, 192, 384 respectively.
> Our method does not only propose structured attention but, for instance, it also includes a probabilistic scheme (i.e. AG-CRF). The table in Fig. 5, in addition to the ablation, is meant to show a comparison between DANet and a version of VISTA-Net in single scale in order to have a more fair comparison. Regarding the computational complexity, the impact of using structured attention can be now seen in Table 6, where we report various measures to assess the additional computational requirements of VISTA-Net. We revised the manuscript fixing typos.

---

### Official Review · AnonReviewer4 · 2020-10-27
**a probabilistic solution to jointly estimate spatial attention maps and channel attention vectors**

**Rating:** 6
**Confidence:** 4

**Review:**

Overview

This paper carefully designs a structured attention network incorporating with a variational solution, to benefit the dense pixel-wise prediction via inferring the latent attention gate between spatial and channel-level features. Based on the suppose of T rank-1 attention tensors, the proposed structured attention network performs a CRF formulation with latent gating variables.

Strengths

1.	Experiments are conducted on various dataset KITTI and NYU-v2 for depth estimation, PASCAL and Cityscapes for semantic segmentation, ScanNet for surface normal prediction. All the results demonstrate the outperformance.

2.	The motivation is clear. Jointly estimating spatial attention maps and channel attention vectors with a probabilistic framework.

Weakness

1.	The contribution is limited. Just a combination of two existing works, e.g., CRF-based models for multi-scale attention estimation and DANet for spatial and channel-wise attention. Why the T rank attention tensors are advanced than full-rank ones? How to guarantee the T channels optimal than the possible C?

2.	The generation of structured latent attention gate is confused. Compared with the work (Xu et al., 2017a), which illustrates the attention-gated CRFs with a clear graph, Figure 1 confuses me. Where is the gate? How to produce a probabilistically enhanced feature map?

3.	The model is complex with four sets of latent variables (Z, M, V, and K) to be inferred. It is hard to be reproduced.

Questions:

1.	How to choose the parameter T? Ablation study might be added to demonstrate the difference with different T.

2.	In the inference stage, there are Z-step, M-step, V-step, and K-step. How to deal with so many variables to obtain the optimal performance? Is there any dependence on data size?

3.	For experiments of depth estimation, why the proposed method obtains optimal metrics but rel?

---

> ### Author Response · Authors · 2020-11-14
> **Response to Reviewer 3**
>
> We thank the reviewer for the positive feedback. Regarding the novelty of the method, we would like to point our answers to R1 for the differences w.r.t. Xu et al. (2017) and to R2 for the differences w.r.t. Fu et al. (2019). We have drawn new diagrams to better explain the relationship between the hidden variables, and the role played by the structured attention tensor, see the new Figure 2. Indeed, the inference procedure is complex, but the code (already submitted in the supplementary material) will be made publicly available, therefore making the method fully reproducible. Together with the generic inference code, we provide scripts to train and test VISTA-Net for all the three tasks considered in the paper. Regarding how to choose the value of T, we provided an ablation study on that matter Figure 4 (right) of the initial submission (now Figure 5). In addition, we also provide some qualitative results in Figure 3 and in Figure 8 of the Appendix. Regarding the computational complexity of the method, we added Table 6 in the appendix, to evaluate the additional computational time required for running VISTA-Net, and we did so for various values of T. The difference in the metric "rel" in Table 2 between Yin et al. (2019) and VISTA-Net is 0.003, which is really small. However, the advantage of VISTA-Net over Yin et al. (2019) in other challenging metrics such as "rms" and "Accuracy $<$ 0.125" is significantly higher. Notice that these metrics capture sligthly different phenomenon, since in "rel" normalisation w.r.t. to ground truth is introduced.

---

### Official Review · AnonReviewer3 · 2020-10-28
**review 2893**

**Rating:** 6
**Confidence:** 4

**Review:**

This paper proposes a unified method to combine spatial and channel attention in a probabilistic framework, so that spatial and channel attention weights and probabilistic variables can be jointly optimized. The proposed method is incoporated in the proposed VISTA-Net to achieve state of the art performance in two dense pixel-wise prediction tasks: monocular depth prediction, semantic segmentation.

Overall this work is well motivated and organized in a good shape, so that I am on the positive side. I have some concerns as listed below.

1. My main concern is that this paper combines ideas of (Fu et al., 2019) and (Xu etal., 2017a), i.e., combining spatial and channel attention from (Fu et al., 2019) and Attention-Gated CRFs from (Xu etal., 2017a).

2. In Eq.1, the attention tensor is limited to T rank. However, it is not clear to me how this can faciliate the inference in Eq.9.

3. In Eq.2, this paper proposes to additionally model CRF kernels. However, it is not explained why this is necessary. This is important as it is listed as a difference to existing method (Xu etal., 2017a).

4. The approximation used in Eq.3 needs more details or reference to support.


Minor issues.
1. There are a lot of symbols in section 2. It would be better to draw a figure to include these symbols, while illustrating how the CRF formulation is incorporated in the attention mechanism.
2. The joint learning in section 2.2 can be formed in a algorithm procedure, so that others could see it more clearly.

---

> ### Author Response · Authors · 2020-11-14
> **Response to Reviewer 2**
>
> We discussed the main differences between our approach and (Xu et al., 2017a) in the answer to R1. Regarding to the difference between VISTA-Net and (Fu et al., 2019) we clearly illustrated it in Fig.1: with respect to DA-Net (Fig.1.c) we jointly model the spatial- and channel-wise attention considering a structured tensor (Fig.1.d). More precisely, in (Fu et al., 2019) two full-rank tensors are estimated independently, and the final attention tensor is computed by an element-wise product. Although two different tensors are computed, the final output has no structure in it. Very differently, we conceived an attention mechanism that is estimating T rank-1 tensors added up to a final tensor with rank T. To do so, we estimate T different spatial attention maps and channel attention vectors and consider the tensor product between the T (attention map, attention vector) pairs. This tensor product provides structure to the attention and eases the learning as demonstrated in our experiments. This is now graphically shown in the new Figure 2, and the accompanying text before Section 2.2. Regarding the modeling of CRF kernels, we believe learning the kernels is important because it allows the CRF to weight the information flow depending on the content of the rather than keeping the same weights for all images. This is now explained in the second paragraph of Section 2. Regarding the variational factorization: considering approximate posterior distributions that are separable in the various kinds of variables involved in inference is standard practice in variational inference. As suggested we have also added the algorithm box (Algorithm 1) and a graphical structure (Fig.14) referring to the VISTA-Net inference network in the supplementary materials.

---

### Official Review · AnonReviewer1 · 2020-10-29
**Recommendation to Reject**

**Rating:** 5
**Confidence:** 4

**Review:**

##########################################################################

Summary:

This paper proposes the VarIational STructured Attention networks (VISTA-Net), which improves pervious SOTA models for dense pixel-wise prediction tasks. The proposed VISTA-Net is featured by two aspects: 1) A new structured attention is proposed, which is able to jointly model spatial-level and channel-level dependencies; 2) It incorporates the proposed structured attention with a CRF-like inference framework, which allows the probabilistic inference. Experimental studies are conducted on monocular depth estimation and semantic image segmentation, showing improved performances of VISTA-Net consistently.

##########################################################################

Reasons for score:

Overall, I vote for rejection. My major concerns lie in three aspects, as detailed in Cons below: 1) This work is highly similar to Xu et al. (2017a) in terms of both methods and presentations. The difference is not significant; 2) While the presentation mainly follows Xu et al. (2017a), it needs some improvement; 3) The experimental studies lack more detailed analysis on the proposed method.

##########################################################################

Pros:

1. The work is well-motivated. The aim of the proposed method sounds natural to me.

2. I like the ablation studies. But they could be performed on at least one more dataset.

##########################################################################

Cons:

1. This work is highly similar to Xu et al. (2017a) in terms of both methods and presentations. The difference is not significant.
Method-wise, as discussed in Related Work, the difference only lies in that VISTA-Net takes channel-level dependencies into consideration. First, this means that "Moreover, we integrate the estimation of the attention within a probabilistic framework" (quoted from abstract) is not a novel contribution. Second, considering channel-level dependencies in attention has limited novelty. As discussed in Related Work, multiple studies have explored several ways. In addition, a key step in the proposed method is Equation (1), where the tensor multiplication operator is not explained. In my understanding, it should be the outer product, or more generally, Kronecker product. Missing the clear definition of this operator hinders the clarity in describing the proposed method.
Presentation-wise, the similarity is even higher. The entire section 2 follows the exact organization of section 2 in Xu et al. (2017a). By comparing the equations and presentations, it's more convincing that the novelty of this work is quite limited.

2. While the presentation mainly follows Xu et al. (2017a), it needs some improvement. First, the same notations as in Xu et al. (2017a) are used. However, some key things are not well explained. For example, the set of hidden variables $z_s$ corresponding to $f_s$ comes out from nowhere. I have to resort to Xu et al. (2017a) to know why we need $z_s$. Second, as mentioned above, the key steps like Equation (1) lack clear explanations.

3. The experimental studies lack more detailed analysis on the proposed method. It would be more meaningful to visualize those attention maps/gates instead of dense prediction results.

##########################################################################

Questions during rebuttal period:

Please address and clarify the cons above

##########################################################################

Comments after the rebuttal period:

Pros:

First, it is totally acceptable to follow the notations and organization of Xu et al. (2017a), as long as the statements are clear and self-contained. The original submission failed on providing key details. The authors have made revisions to address this concern. Thanks!

Second, I appreciate the extra experimental results and visualizations.

Cons:

However, the authors' responses do not fully address my concerns about the novelty, especially method-wise. I will raise my score to 5, but still recommend rejection.

---

> ### Author Response · Authors · 2020-11-14
> **Response to Reviewer 1**
>
> We thank R1 for the feedback provided, and for acknowledging the motivation of our work as well as the intuitiveness of the method we propose. Regarding the ablation studies, despite the fact that is not common practice to ablate in various dataset, we will do our best to run the necessary experiments and submit them before the rebuttal deadline. We would like to comment on the similarities with Xu et al. (2017a). Regarding the presentation, we used the same notation and flow on purpose, because of two main reasons: clarity and transparency. Indeed, for a reader interested in pixel-level prediction, papers sharing the notation are considerably easier to read. In addition, it also allows to present more clearly the novelty of our approach w.r.t. to  Xu et al. (2017a) in a very transparent way. In this regard, we disagree with the reviewer statement "The difference is not significant.'' This paper is different w.r.t. Xu et al. (2017a) in many aspects. First of all, the main motivation of VISTA-Net is to structure the attention tensor. This is radically different from Xu et al. (2017a) where the authors focus in processing multi-scale features. From the methodological point of view, in VISTA-Net we propose to structure the attention tensor via a summation of several rank-1 tensors. Furthermore, we include the inference of these tensors, and therefore of the structured attention, within a probabilistic CRF. Very importantly, from an application point of view, we demonstrated the interest of our approach on various tasks and datasets, demonstrating the generic interest and usability of VISTA-Net. In the revised version of the manuscript, we have clarified the formalisation of the method around equation (1) and the need for hidden variables in the 3rd paragraph of Section 2. In addition, we display the resulting attention maps of a few samples of the KITTI dataset in Fig. 13 of the revised version.

---

### Author Response · Authors · 2020-11-14
**Summary of changes**

We thank the reviewers for the precious suggestions, here we try to address their comments punctually. We have also uploaded a revised version of the paper, including the modifications highlighted in color to be easily traceable. We will remove colored text right before the deadline.
Summary of the major modifications:
1. we clarified the formalization of the method around equation (1) explaining the need for hidden variables in the 3rd paragraph of Section 2;
2. we show the resulting attention maps on a few images of KITTI dataset in Fig. 13
3. we have added a new Figure 2 and an algorithm box (in the supplementary materials) to drive the reader into the comprehension of the architecture of VISTA-Net.

---

### Decision · Program_Chairs · 2021-01-07
**Final Decision**

**Decision:**

Reject

**Comment:**

Reviewers were concerned with the novelty, although appreciated sota results in extensive experiments.